# GENERATIVE STRESNET FOR CRIME PREDICTION

**Tran Phong & Hoong Chuin Lau** *
School of Computing and Information Systems
Singapore Management University
80 Stamford Rd, Singapore

## ABSTRACT

In this work, we combine STResnet (Zhang et al., 2017) with VAE Kingma &
Welling (2013) to generate crime distribution. The outputs can be used for down-
stream tasks such as patrol deployment planning Chase et al. (2021).

## 1 INTRODUCTION

Predicting crime is a critical aspect of public safety. However, the complexity and unpredictability
of crime patterns make this task challenging. This paper introduces a novel approach that incorpo-
rates stochasticity into spatio-temporal crime prediction. Our method, which combines a Variational
Autoencoder (VAE) with Spatio-Temporal Residual Networks (STResnet), generates a range of po-
tential crime scenarios. These scenarios can be utilized in downstream planning tasks, such as patrol
deployment planning, as outlined in Chase et al. (2021).

## 2 RELATED WORKS

Several deep learning methods have been proposed for crime prediction. Huang et al. (2018) used
a hierarchical GRU with attention to indicate binary pixels in images, where each pixel represents
a map region. Ye et al. (2021) employed an inception-residual CNN to predict crime probabilities
in discretized grids. Wang et al. (2019) utilized STResnet, as proposed by Zhang et al. (2017), for
crime prediction. However, a common limitation of above methods is the lack of stochastic outputs.
This restricts their utility in deployment planning tasks, such as those described in Chase et al.
(2021), which require multiple scenario inputs to generate robust plans. To address this limitation,
our paper presents a novel approach that integrates stochasticity into crime predictions.

## 3 GENERATIVE ST-RESNET ARCHITECTURE

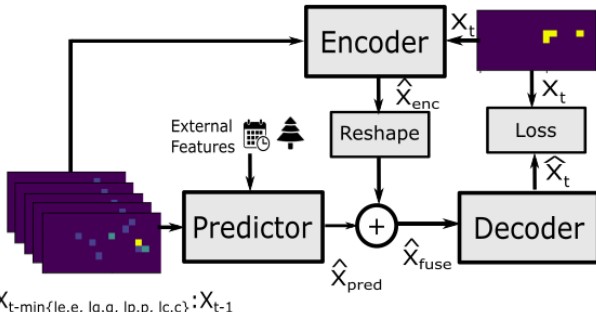

Figure 1: Architecture Diagram of Generative ST-ResNet

Our Generative ST-ResNet architecture, as depicted in Figure 1, is designed to process a time se-
quence of images, where each image represents a crime intensity map. We first describe how maps

---

*Corresponding Author hclau@smu.edu.sg.

are created, then we explain the roles of the architecture's three primary components: the *Predictor*, *Encoder*, and *Decoder*.

To construct the crime intensity maps, we first divide a specific region (the area where we aim to predict crime timings and locations) into an $M \times N$ grid map. The size of each grid square is determined by a parameter we refer to as *resolution*. We then aggregate crime incidents within a given *interval* and use the count as the grid's intensity.

From the sequence of these generated images, we create four distinct sequences or fragments: *instant*, *closeness*, *period*, and *trend*. The lengths of these fragments are represented as $l_e$, $l_c$, $l_p$, and $l_q$, and they are created based on specific sampling rates, denoted by $e$, $c$, $p$, and $q$, respectively. We use the following equations to segment the data in each window into these four sequences:

$$S_e = [X_{t-l_e.e}, X_{t-(l_e-1).c}, ..., X_{t-e}]$$
$$S_c = [X_{t-l_c.c}, X_{t-(l_c-1).c}, ..., X_{t-c}]$$
$$S_p = [X_{t-l_p.p}, X_{t-(l_p-1).p}, ..., X_{t-p}]$$
$$S_q = [X_{t-l_q.c}, X_{t-(l_q-1).q}, ..., X_{t-q}]$$

The window length is determined by $min\{l_e.e, l_c.c, l_p.p, l_q.q\}$ plus 1 to include the target image $X_t$.

The *Predictor* block of the architecture processes the *closeness*, *period*, and *trend* sequences, which model the temporal properties of the data. This block also receives cyclical data related to the timing of incidents (e.g., whether they occurred on a weekend, weekday, or holiday).

The *Encoder* block takes as input the instant sequence, which represents the *noise* in the data, and the *target*. The *Encoder* block outputs a random latent variable that captures both the noise in the data and the uncertainty of the prediction.

Finally, the *Decoder* block receives inputs from both the *Predictor* and *Encoder* blocks. This block contains a series of transposed convolutional networks designed to match the size of the output.

More details of three blocks are found in the Appendix.

## 4 EXPERIMENTS AND RESULTS

Our study utilized a year's worth of crime incident data from Singapore in 2020 to train the neural network, and subsequently predicted crimes for the first half of 2021. The data, which included start time, latitude, and longitude for each incident, spanned geographically from $1.2101°$ to $1.4707°$ in latitude and from $103.6056°$ to $104.0436°$ in longitude.

The network was trained using Mean Squared Error (MSE) Loss with the following parameters: $resolution = 0.015, interval = 60$ minutes, $l_e = 24, e = 1, l_c = 12, c = 1$ (hourly closeness and instant), $l_p = 3, p = 24$ (daily period), $l_t = 1, t = 168$ (weekly trend).

To evaluate the performance, we compared our method (STResnetVAE) with STResnet Zhang et al. (2017) and GAN Goodfellow et al. (2020) using the F1 score. The intensity output was converted into latitude, longitude, and timestamp. In cases where more than one incident occurred at the same pixel, a time within the timestamp duration was randomly selected. Predicted incidents were then matched with ground-truth incidents using linear assignment under various distances and durations.

Table 1: F1 score

| Parameters | | Results | | |
|---|---|---|---|---|
| DISTANCE | DURATION | GAN | STResnet | STResnetVAE (Ours) |
| 1000 m | 10 mins | 0.15 | **0.167** | 0.150 |
| 2000 m | 30 mins | 0.40 | **0.598** | 0.596 |
| 2000 m | 60 mins | 0.43 | 0.749 | **0.755** |

From the Table 1, we see that STResnetVAE performed well in predicting incidents when the distance and duration were large. However, it was less effective for smaller instances. Despite this, STResnetVAE demonstrated comparable performance to STResnet, with the added advantage of being able to predict the distribution of outputs. This feature provides the downstream planning task with the ability to optimize based on diverse predicted outcomes.

URM STATEMENT

The authors acknowledge that one key author of this work meets the URM criteria of ICLR 2023 Tiny Papers Track.

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

A APPENDIX

A.1 STRUCTURES OF PREDICTOR BLOCK

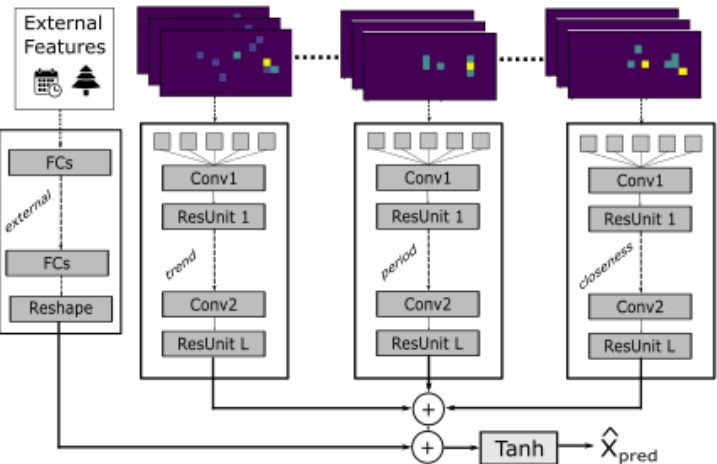

Figure 2: Predictor Block of Generative ST-ResNet

The *Predictor* block processes three temporal sequences: *closeness*, *period*, and *trend*. Each sequence is transformed into a 3D Tensor by concatenating the intensity maps along the last axis (channels). These tensors, along with external features, are propagated through the ST-ResNet architecture to generate the output, denoted as $\hat{X}_{pred}$ as in Figure 2

## A.2 STRUCTURES OF ENCODER BLOCK

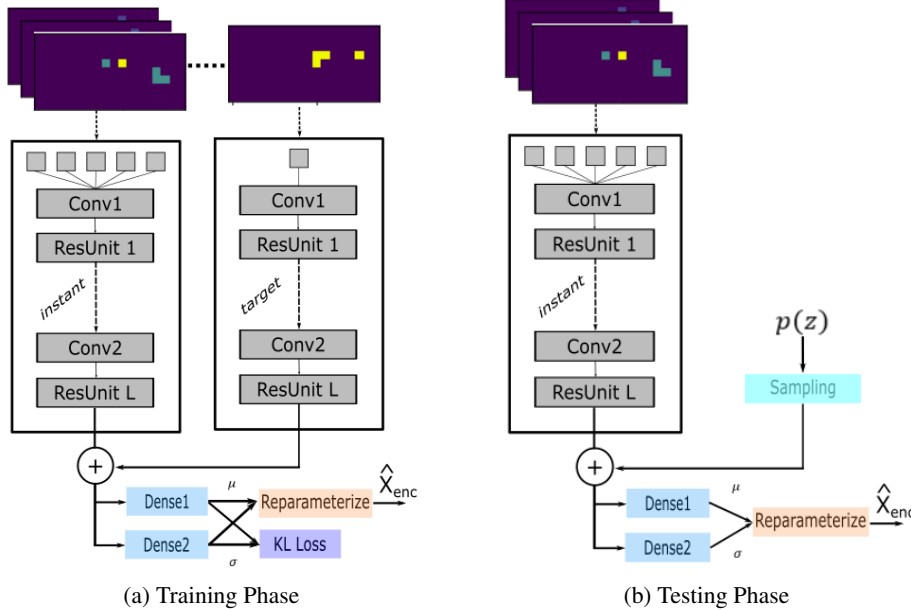

(a) Training Phase          (b) Testing Phase

Figure 3: Encoder Block. (a) Training Phase when we incorporate the target to learn the uncertainty; (b) Testing phase when we sample a prior distribution

During the training phase Figure 3a, the *Encoder* block receives two types of inputs: sequences of instant inputs and targets. These sequences are processed in the same way as in the *Predictor* block: they are concatenated along the channels axis to form a single tensor. This tensor and the target are then propagated through a separate network (with the same architecture) to produce two outputs that are parametrically fused:

$$X = W_e \circ X_e^{(L)} + W_{target} \circ X_{target}^{(L)} \tag{1}$$

where $\circ$ is Hadamard product, $W_e$, $W_{target}$ are trainable parameters. The fused output is then passed through two MLP layers to output mean $\mu$ and variance $\sigma$ to use in KL Loss and sampling $\hat{X}_{enc}$.

In the Testing phase Figure 3b, only sequence of instant inputs are propagated through the network for sampling $\hat{X}_{enc}$

## A.3 STRUCTURES OF DECODER BLOCK

The input to the *Decoder* block is $\hat{X}_{fuse}$, which is the addition of $\hat{X}_{pred}$ from the *Predictor* block and $\hat{X}_{enc}$ from the *Encoder* block through the *Reshape* module. Decoder consists of a series of transposed convolutional neural network layers. We do not use skip connection. The output of *Decoder* block is our final prediction.

Our Generative ST-ResNet can be trained to predict $X_t$ from the sequences of previous inputs by joinlty minimizing two losses. The first loss is the mean squared error (reconstruction loss) between the predictied output and target output:

$$\mathcal{L}(\theta) = ||X_t - \hat{X}_t||_2^2 \tag{2}$$

where $\theta$ is all trainable parameters of all blocks.

The second loss is the KL Divergence loss between the output of *Encoder* and a prior distribution (we choose normal distribution $\mathcal{N}(0, 1)$):

$$\mathcal{L}(\phi, \theta, x) = D_{KL}(q_\phi(z|x)||p_\theta(z)) \tag{3}$$

