# OpenReview forum: "Generative STResnet for Crime Prediction"
_ICLR.cc/2023/TinyPapers — Submitted to Tiny Papers @ ICLR 2023_

### Official Review · Reviewer_FMUb · 2023-03-22

**Confidence:** 3

**Summary Of Contributions:**

Authors propose an approach combining existing STResnet and VAE for crime prediction. The proposed approach shows promising results in long distance prediction and on par performance in short distance prediction with existing methods.

**Rating:**

Clear, Correct, and Reproducible (CCR): a submission which meets the reviewing criteria

**Strengths And Weaknesses:**

* Strengths
    * The proposed approach is simple and easy to implement.
    * The proposed approach shows promising results in long distance prediction and on par performance in short distance prediction with existing methods.
    * Comparisons with GAN and STResnet are provided.
* Weaknesses
    * An extensive ablation study is not provided.

**Suggested Changes:**

I think the proposed approach is simple and easy to implement, an ablation study will justify the effectiveness of the proposed approach. Still, I think the proposed approach is a good starting point for future work. I recommend the paper for discussion at ICLR Tiny Papers.

---

> ### Author Response · Authors · 2023-05-23
> **Thank you for your comments**
>
> We appreciate your thoughtful and constructive feedback on our paper. We're glad to hear that you found our approach simple, easy to implement, and promising in terms of results.
>
> We acknowledge your concern regarding the lack of an extensive ablation study, and we agree that such an investigation would further justify the effectiveness of our proposed approach. At this point, we do not have access to data anymore for doing an ablation study as we closed the project with third party which provided the crime data for us. However, we acknowledge this drawback and we hope to incorporate ablation study in future papers
>
> We also thank you for recognizing the comparative evaluation with GAN and STResnet, and for acknowledging our work as a good starting point for future work in this field. We rewrite some parts in the paper so that each part has a smooth transition and easy to understand.
>
> Once again, we truly thank you for your valuable review. We hope this is a good initial point for us to develop further advanced works.

---

### Official Review · Reviewer_2bpJ · 2023-04-01

**Confidence:** 3

**Summary Of Contributions:**

The main claim of the paper is that a previously published method for crime prediction can be augmented to produce not only a single spatiotemporal prediction, but a probability distribution over all possible spatiotemporal values. This is achieved by adding a VAE to STResnet.

**Rating:**

Great Start (GS): a submission which meets some of the reviewing criteria but has room for improvement

**Strengths And Weaknesses:**

The major weakness of this paper is its lack of clarity. In particular, details on the problem and the solution are sparse and/or confusing throughout the paper. Furthermore, although the main contribution of the paper is obtaining a probability distribution over the predictions as compared to prior solutions, there is no discussion or showcasing of this in the results. F1 scores are the only evaluation metric presented which do not surpass prior work substantially, making it difficult to judge the merits and contribution of the current work.

Since the subject matter of the of the paper is crime prediction, any ethical implications of the shortcomings of the model are important to consider. There doesn't seem to be any discussion of this issue in the paper.

**Suggested Changes:**

The introduction must be expanded to frame the problem and motivate the solution better. The Figures are a waste of space if they do not have proper captioning to aid in understanding the approach.

There is no high-level description of the problem or the solution. For example, although it is mentioned that past crime data are used as inputs to the model, it is unclear what aspects of the data the Resnet/VAE architecture takes advantage of to predict location and timing of crime.

The descriptions of the approach are also very confusing. The elements of the solution must be clearly described. It must be explained why things are done and how they aid in achieving a better solution. For example, what is the slicing operation? What is the "target"? What are the four "fragments": instant, closeness, period, and trend? Why are they useful?

For evaluation, a comparison with STResnet is appropriate as this work builds directly on it. However, using vanilla GANs for comparisons is not informative. Why not use a VAE for this? This would make it easier to directly compare individual components (STResnet and VAE) to the combination of the two (STResnetVAE) for a fair assessment of whether or not the components, when combined, have synergistic success.

---

> ### Author Response · Authors · 2023-05-23
> **Thank you for your comments**
>
> We sincerely thank you for your detailed review and helpful suggestion on our paper. We are glad that you have taken your time to deeply analyze our work. We truly appreciate your effort to help us understand our paper's drawbacks. We cite the reviewer's comments and address them below.
>
> > The introduction must be expanded to frame the problem and motivate the solution better. The Figures are a waste of space if they do not have proper captioning to aid in understanding the approach.
>
> Firstly, we have already expanded our introduction to better frame the problem and motivate the solution. We agree that the figures can be more meaningful with proper captioning. In the revised version, we provided comprehensive captions to better convey the intent of each figure and to improve the overall understanding of our approach.
>
> > There is no high-level description of the problem or the solution. For example, although it is mentioned that past crime data are used as inputs to the model, it is unclear what aspects of the data the Resnet/VAE architecture takes advantage of to predict location and timing of crime.
>
> Secondly, regarding the high-level description, we provided more clarity in our revised version. We elaborated on how the Resnet/VAE architecture utilizes crime data for prediction.
>
> > The descriptions of the approach are also very confusing. The elements of the solution must be clearly described. It must be explained why things are done and how they aid in achieving a better solution. For example, what is the slicing operation? What is the "target"? What are the four "fragments": instant, closeness, period, and trend? Why are they useful?
>
> Thirdly, we also explaind terms like 'slicing operation', 'target', and the 'four fragments', and their significance in our approach. We sincerely thank you reviewer's for this important feedback as it indeed makes reader easy to understand the concepts of our paper.
>
> > For evaluation, a comparison with STResnet is appropriate as this work builds directly on it. However, using vanilla GANs for comparisons is not informative. Why not use a VAE for this? This would make it easier to directly compare individual components (STResnet and VAE) to the combination of the two (STResnetVAE) for a fair assessment of whether or not the components, when combined, have synergistic success.
>
> Fourthly, we acknowledge that we should use VAE for comparison. At this point, we do not have access to the data provided by a third party (as this crime data is sensitive), we are not able to do evaluation for VAE. But we truly thank you for your suggestion and we take this as experience to do related comparison in our future works.
>
> Finally, your feedback is greatly appreciated and has given us valuable insights for improving our paper. Thank you once again for your time and expertise.

---

### Comment · Area_Chair_ANde · 2023-06-06
**Archive Criteria Check.**

This work meets the threshold for archival, contents the URM statement and is deanonymized

---

### Meta-Review · Area_Chair_ANde · 2023-04-08

**Recommendation:** Invite to archive
**Confidence:** 4

**Metareview:**

Pros and cons summarized by the comments from the reviewers:

Strength:

1. The proposed approach is simple and easy to implement.
2. The method shows promising results in long-distance prediction and on-par performance in short-distance prediction compared to existing methods.
3. Comparisons with GAN and STResnet are provided.

Weakness:

1. The paper lacks clarity, especially in explaining the problem and solution.
2. There is no discussion on the probability distribution over predictions, which is the main contribution.
3. Limited evaluation metrics are presented, making it difficult to judge the merits and contribution.
4. The paper does not discuss any ethical implications of the model's shortcomings.
5. An extensive ablation study is not provided.


**Summary:**

The paper proposes an approach that combines STResnet and VAE for crime prediction, offering probability distributions over spatiotemporal values and showing promising results in long-distance predictions.

**Comments And Feedback To The Authors:**

You proposed a novel approach for crime prediction by combining STResnet and VAE. The paper meets most criteria, except for clarity, as mentioned by one of the reviewers. I highly recommend working on the writing and structure of the paper, which includes better writing, clearer definitions of each element, more background introduction, and a description of the high-level idea of your method.

**Reason For Not Giving A Higher Recommendation:**

The authors proposed a novel approach for crime prediction by combining STResnet and VAE. The method is simple, easy to implement, and the comparison experiment against STResnet and GAN shows promising results. However, as mentioned by the reviewers, the overall writing quality of the paper needs significant improvement. The authors are encouraged to work on enhancing the clarity of the paper.

**Reason For Not Giving A Lower Recommendation:**

I believe the authors have achieved all the criterions except for clarity, which would be achieved a revision. A significant revision is not necessary.

---

### Decision · Program_Chairs · 2023-04-08

Invite to archive